# Cryo-EM structure of the potassium-chloride cotransporter KCC4 in lipid nanodiscs

Michelle S Reid[1,2], David M Kern[1,2], Stephen Graf Brohawn[1,2]*

[1]Department of Molecular and Cell Biology, University of California Berkeley, Berkeley, United States; [2]Helen Wills Neuroscience Institute, University of California Berkeley, Berkeley, United States

**Abstract** Cation-chloride-cotransporters (CCCs) catalyze transport of Cl- with K+ and/or Na+ across cellular membranes. CCCs play roles in cellular volume regulation, neural development and function, audition, regulation of blood pressure, and renal function. CCCs are targets of clinically important drugs including loop diuretics and their disruption has been implicated in pathophysiology including epilepsy, hearing loss, and the genetic disorders Andermann, Gitelman, and Bartter syndromes. Here we present the structure of a CCC, the *Mus musculus* K+-Cl- cotransporter (KCC) KCC4, in lipid nanodiscs determined by cryo-EM. The structure, captured in an inside-open conformation, reveals the architecture of KCCs including an extracellular domain poised to regulate transport activity through an outer gate. We identify binding sites for substrate K+ and Cl- ions, demonstrate the importance of key coordinating residues for transporter activity, and provide a structural explanation for varied substrate specificity and ion transport ratio among CCCs. These results provide mechanistic insight into the function and regulation of a physiologically important transporter family.

*For correspondence:
brohawn@berkeley.edu

Competing interests: The authors declare that no competing interests exist.

## Introduction

CCCs in mammals include the potassium-chloride cotransporters KCC1-4, the sodium-potassium-chloride cotransporters NKCC1-2, the sodium-chloride cotransporter NCC, and CCC8-9 (*Figure 2—figure supplement 1*; *Arroyo et al., 2013*; *Marcoux et al., 2017*; *Gamba, 2005*). First characterized as modulators of red blood cell volume (*Dunham et al., 1980*; *Lauf and Theg, 1980*), CCCs are now appreciated to play critical roles in cellular volume regulation, modulation of neuronal excitability, renal function, auditory system function, transepithelial transport, and blood pressure regulation (*Marcoux et al., 2017*; *Gamba, 2005*; *Singhvi et al., 2016*). CCCs are targets of drugs including the thiazide and loop diuretics hydrochlorothiazide, furosemide, and bumetanide and their disruption is associated with congenital hydrocephaly, epilepsy, hearing loss, Andermann syndrome, Gitelman syndrome, and Bartter syndrome (*Gamba, 2005*; *Jin et al., 2019*; *Kahle et al., 2015*).

KCCs are important for K+ and Cl- homeostasis, including in establishing low neuronal cytoplasmic Cl- concentrations critical for inhibitory neurotransmission, and in volume regulation in many cell types (*Marcoux et al., 2017*; *Bergeron et al., 2003*; *Mount et al., 1999*; *Karadsheh et al., 2004*). Among KCCs, KCC4 is most strongly activated by cell swelling and high internal [Cl-] and is uniquely active in acidic external environments (*Marcoux et al., 2017*; *Bergeron et al., 2003*). KCC4 is expressed in tissues including the heart, nervous system, kidney, and inner ear and mice lacking KCC4 display progressive deafness and renal tubular acidosis (*Marcoux et al., 2017*; *Mount et al., 1999*; *Karadsheh et al., 2004*; *Boettger et al., 2002*). Hearing loss in these animals is due to disrupted K+ recycling by Dieter's cells in the cochlea and hair cell excitotoxicity, while renal tubular

acidosis is due to impaired $Cl^-$ recycling by $\alpha$-intercalated cells in the kidney distal nephron (*Boettger et al., 2002*).

CCCs display varied substrate specificity and transport stoichiometry despite sharing a common amino acid-polyamine-organocation (APC) superfamily fold (*Payne, 2012*; *Hartmann and Nothwang, 2014*; *Shi, 2013*). KCCs cotransport $K^+:Cl^-$ in a 1:1 ratio, NKCCs cotransport $1K^+:1Na^+:2Cl^-$, and NCCs cotransport $1Na^+:1Cl^-$. One consequence of this difference is that under typical conditions (with $[K^+]_{in}:[K^+]_{out} > [Cl^-]_{out}:[Cl^-]_{in}$), transport by KCCs is outwardly directed while transport by NKCCs/NCCs is directed into cells (*Marcoux et al., 2017*; *Gamba, 2005*).

CCCs have two distinctive elaborations on the APC fold. First, the scaffold is followed by a C-terminal domain (CTD) important for regulating expression, trafficking, and activity including through phosphorylation or dephosphorylation of CTD sites in response to cell swelling (*Rinehart et al., 2009*; *Bergeron et al., 2009*; *Frenette-Cotton et al., 2017*; *Melo et al., 2013*). Second, CCCs contain a 'long extracellular loop' with predicted disulfide bonds and glycosylation sites that differs in position and structure between CCCs; it is formed by the region between TM5-TM6 in KCCs and between TM7-TM8 in NKCCs (*Hartmann and Nothwang, 2014*; *Hartmann et al., 2010*).

KCCs are present as monomers and dimers in cells and modulation of quaternary state has been implicated in transporter regulation. A shift from monomeric to dimeric KCC2 during development coincides with an increase in its activity that results in chloride extrusion from neurons (the excitatory-to-inhibitory GABA switch) (*Rivera et al., 1999*; *Blaesse et al., 2006*; *Puskarjov et al., 2012*). Homodimerization is thought to be largely mediated through CTD interactions, as observed in the recent cryo-EM structure of NKCC1 (*Chew et al., 2019*), and calpain-mediated proteolysis of the KCC2 CTD is associated with a decrease in transporter activity (*Puskarjov et al., 2012*). In addition to self-associating, KCCs heterodimerize with other CCCs and interact with other membrane proteins including ion channels (*Blaesse et al., 2006*; *Simard et al., 2007*).

Here we report the structure of *Mus musculus* KCC4 in lipid nanodiscs determined by cryo-EM. The structure reveals unique features of KCCs and, together with functional characterization of structure-based mutants, provides insight into the basis for ion binding, transport, and regulation of KCC4 activity.

## Results

### Structure of KCC4 in lipid nanodiscs

*Mus musculus* KCC4 was heterologously expressed in *Spodoptera frugiperda* (Sf9) insect cells for purification and structure determination (*Figure 2—figure supplement 2*). To assess the activity of KCC4 in these cells, we utilized an assay that depends on the ability of KCCs to transport $Tl^+$ in addition to $K^+$ (*Zhang et al., 2010*). In cells loaded with the $Tl^+$-sensitive fluorophore FluxOR red, $Tl^+$ uptake from the extracellular solution results in an increase in fluorescence signal (*Figure 1A*). Cells infected with virus encoding KCC4, but not cells infected with a virus encoding an anion-selective volume-regulated ion channel SWELL1 (*Kern et al., 2019*) or uninfected Sf9 cells, displayed increased fluorescence over time consistent with KCC4 activity (*Figure 1B,C*). No significant difference in activity was observed between N- and C-terminally GFP-tagged mouse KCC4 (*Figure 1B,C*), in contrast to a previous report for KCC2 (*Agez et al., 2017*), and C-terminally tagged KCC4 was used for subsequent study.

We reconstituted KCC4 into lipid nanodiscs in order to study the structure of the transporter in a native-like membrane environment. KCC4 was extracted, purified in detergent, and exchanged into nanodiscs formed by the membrane scaffold protein MSP1D1 and a mixture of phospholipids that approximates the composition of major species in neuronal membranes (2:1:1 molar ratio DOPE:POPC:POPS (2-dioleoyl-sn-glycero-3-phosphoethanolamine:1-palmitoyl-2-oleoyl-sn-glycero-3-phosphocholine:1-palmitoyl-2-oleoyl-sn-glycero-3-phospho-L-serine)) (*Figure 2—figure supplement 2*; *Ingólfsson et al., 2017*; *Ritchie et al., 2009*). KCC4-MSP1D1 particles are similar in size and shape to KCC4 particles in detergent micelles by cryo-EM, but show improved distribution in thin ice which enabled reconstruction to high resolution (*Figure 2—figure supplement 3*).

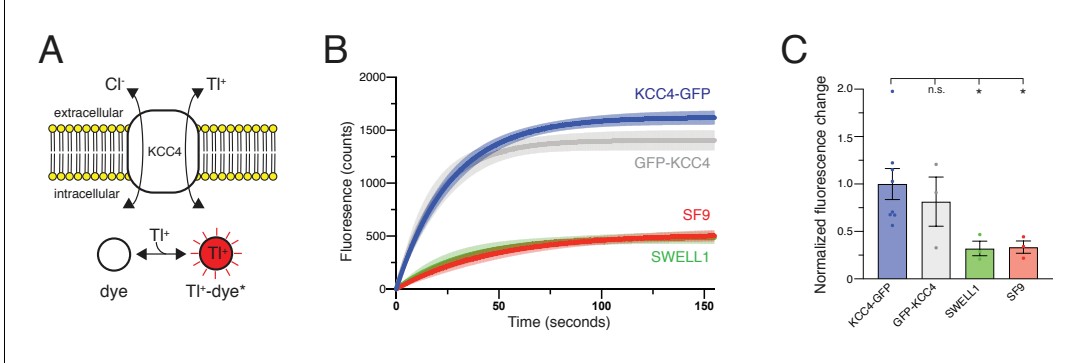

**Figure 1.** Transport activity of mouse KCC4. (**A**) Tl$^+$ uptake assay for KCC4 activity. KCC4 activity in SF9 cells results in Tl$^+$ uptake and increased fluorescence of the Tl$^+$ sensitive dye FluxOR Red. (**B**) Fluorescence values as a function of time for each construct assayed. Lines are global exponential fits to all data with 95% confidence intervals shown for KCC4-GFP (n = 8, blue), GFP-KCC4 (n = 3, gray), SWELL1 (n = 3, green), and uninfected SF9 cells (n = 3 red). (**C**) Quantification of experiments shown in (**B**). Normalized final fluorescence. KCC4-GFP 1.0 ± 0.16 (n = 8); GFP-KCC4 0.76 ± 0.33 (n = 3) SWELL1 0.32 ± 0.08 (n = 3) SF9 0.33 ± 0.06 (n = 3); mean ± SEM, one-way Anova (*p<0.05, n.s. = not significant).

An unmasked reconstruction of KCC4 in nanodiscs is shown in *Figure 2A* contoured to highlight the position of the lipid belt surrounding the transmembrane region. To achieve the highest resolution reconstruction, the nanodisc density was subtracted and particles were subjected to focused classification and subsequent refinement (*Figure 2—figure supplement 4*). The resulting map, at 3.65 Å overall resolution, enabled complete de novo modeling of the transmembrane and extracellular region of KCC4 and includes two partial extracellular glycosylation sites, a bound K$^+$ ion, and a bound Cl$^-$ ion (*Figure 2C,D, Figure 2—figure supplements 5* and *6*).

## Overall architecture

KCC4 is monomeric in the nanodisc structure. Density for the N-terminal region and C-terminal domain (CTD), which together comprise approximately half the expressed protein mass, is not observed in the cryo-EM maps (*Figure 2*). The N-terminal region is weakly conserved, variable in length among CCCs (*Figure 2—figure supplement 1*), and is likewise unresolved in the structures of NKCC1 or KCC1 (*Chew et al., 2019*; *Liu et al., 2019*). We presume it is highly flexible in KCC4. The C-terminal domain, while similarly unresolved in KCC1 (*Liu et al., 2019*), is well conserved, has documented roles in regulation, expression, and trafficking (*Arroyo et al., 2013*; *Marcoux et al., 2017*; *Payne, 2012*; *Hartmann and Nothwang, 2014*; *Rinehart et al., 2009*), and mediates homodimerization of NKCC1 and the Archaean CCC (MaCCC) (*Chew et al., 2019*; *Warmuth et al., 2009*). We found no evidence of proteolytic cleavage of either region. Mass spectrometry of purified KCC4 showed high coverage (47%) and abundance (81% of all KCC4 peptides) for the CTD (*Figure 2—figure supplement 7*). We observe a progressive loss of detailed features and decrease in local resolution in TM11 and TM12 that connect the CTD to the core transmembrane region (*Figure 2—figure supplements 5B* and *6*). Some two-dimensional class averages show a blurred cytoplasmic feature in the position we expect the CTD to emerge (*Figure 2—figure supplement 3B,D*), but attempts to classify distinct conformations of this feature were unsuccessful. We conclude that the monomeric structure reported here corresponds to full-length mouse KCC4 with flexible and/or disordered terminal regions.

The monomeric structure of KCC4 contrasts with recent homodimeric structures of *Danio rerio* NKCC1 (*Chew et al., 2019*) and *H. sapiens* KCC1 (*Liu et al., 2019*), although dimerization of NKCC1 and KCC1 involve completely distinct interfaces (*Figure 3A,C*). Disruption of putative tightly associated KCC4 homodimers during purification or sample preparation was excluded for the following reasons: (i) The portion of KCC4 in an early-eluting broad peak from a sizing column (*Figure 2—figure supplement 2A*) displays nonspecific aggregation by cryo-EM. (ii) KCC4 is monomeric before and after reconstitution in nanodiscs as assessed by cryo-EM (*Figure 2—figure supplements 2* and *3*). (iii) Cross-linking of purified KCC4 was observed only at high concentrations of crosslinker and was reduced when KCC4 was first deglycosylated (*Figure 2—figure supplement 2F*), suggesting some cross-linking in glycosylated KCC4 is from intermolecular glycan-glycan or protein-glycan

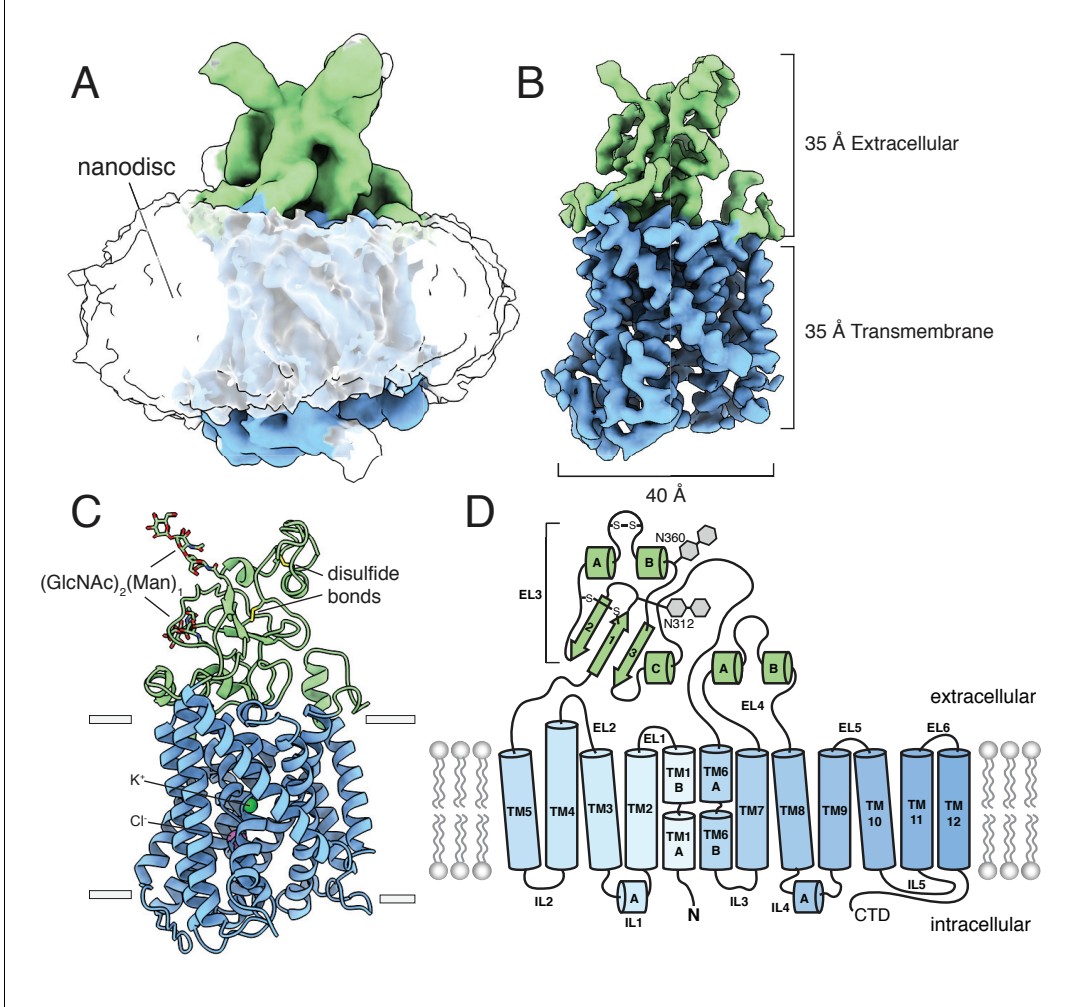

**Figure 2.** Structure of mouse KCC4 in lipid nanodiscs. (**A**) Cryo-EM map from an unmasked refinement viewed from the membrane plane showing the position of nanodisc, transmembrane region (blue), and extracellular region (green). (**B**) Final map, (**C**) corresponding atomic model, and (**D**) cartoon representation of KCC4. In (**C**), bound $K^+$ and $Cl^-$ ions are shown as green and violet spheres, respectively. Two disulfides and two N-linked glycosylation sites are shown as sticks and labeled in the cartoon (a third disulfide between TM2 and TM11 is not visible).

The online version of this article includes the following figure supplement(s) for figure 2:

**Figure supplement 1.** CCC family sequence alignment.
**Figure supplement 2.** Purification and reconstitution of mouse KCC4.
**Figure supplement 3.** Example micrograph and 2D class averages.
**Figure supplement 4.** Cryo-EM processing pipeline for KCC4 in MSP1D1 nanodiscs.
**Figure supplement 5.** Cryo-EM validation.
**Figure supplement 6.** Representative regions of cryo-EM map.
**Figure supplement 7.** Mass spectrometry of purified KCC4.
**Figure supplement 8.** FSEC comparison of KCC4 and KCC1 expressed in different host cells and treated with different detergents.

linkages rather than through transmembrane regions or CTDs (*Chew et al., 2019*). (iv) No substantial differences were observed in the apparent size of KCC4 or KCC1 (assessed by gel filtration) transporters extracted from different expression host cells or treated with different combinations of detergents used in the CCC structure reports to date (*Chew et al., 2019*; *Liu et al., 2019*; *Figure 2—figure supplement 8*).

We asked whether there are functional consequences of putative KCC4 dimerization through interfaces similar to those observed in KCC1 or NKCC1 structures. The KCC1 dimeric interface is mediated predominantly through protein-detergent interactions between TM regions and protein-

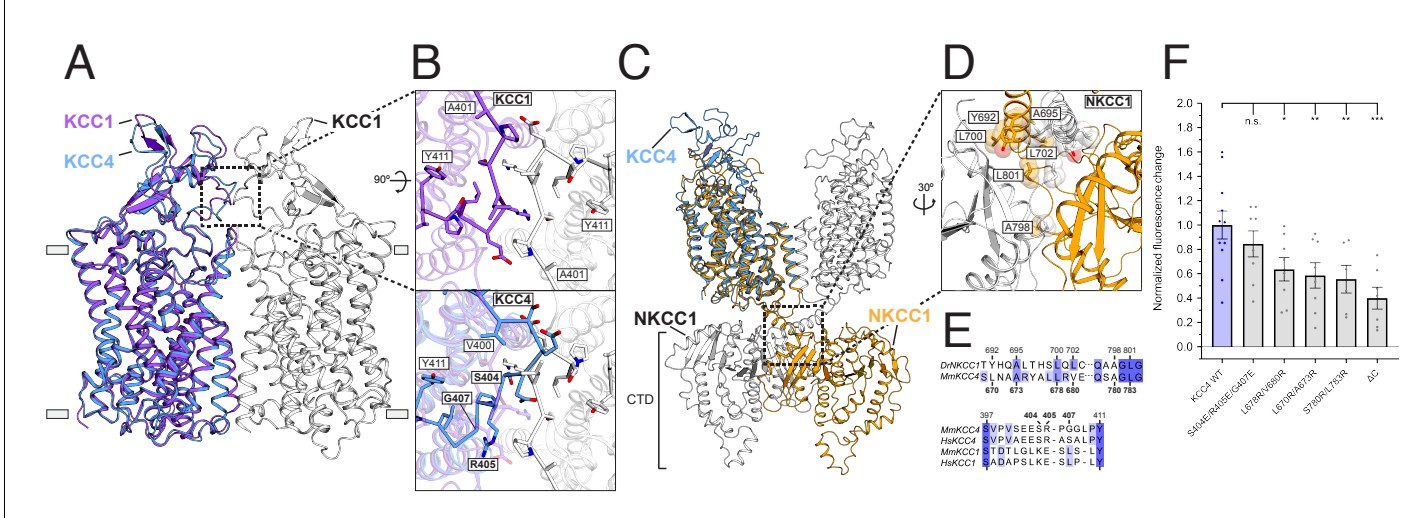

**Figure 3.** Analysis of putative KCC4 dimerization. (**A**) Overlay of monomeric KCC4 (blue) and dimeric KCC1 (protomer one purple, protomer two white, PDB 6KKT) structures viewed from the membrane. (**B**) Magnified top views of the extracellular protein-protein interaction region of KCC1 (dashed box in (**A**)) shown alone with residues in KCC1 labeled (upper) or with KCC4 overlaid (lower) and KCC4 residues labeled. (**C**) Overlay of KCC4 (blue) and dimeric NKCC1 (protomer one orange, protomer two white, PDB 6NPL) structures viewed from the membrane. (**D**) Magnified view of the intracellular protein-protein interaction region of NKCC1 (dashed box in (**C**)). Residues in NKCC1 for which corresponding KCC4 mutations were made are labeled. (**E**) Partial sequence alignments between KCC4 and NKCC1 (above) and human and mouse KCC1 and KCC4 (below) for the regions highlighted in (**B**, **D**). Residues mutated in KCC4 are numbered in bold. (**F**) Normalized activity of KCC4 mutations. Wild-type KCC4 1 ± 0.11 (n = 11); S404E, R405E, G407E 0.85 ± 0.11 (n = 8); L678R, V680R 0.59 ± 0.10 (n = 8); L670R, A673R 0.64 ± 0.10 (n = 8); S780R, L783R 0.56 ± 0.11 (n = 6); KCC4ΔC (1–658 0.40 ± 0.09 (n = 7); mean ± SEM, one-way Anova (*p<0.05, **p<0.01 ***p<0.001).

The online version of this article includes the following figure supplement(s) for figure 3:

**Figure supplement 1.** FSEC comparison of KCC4 mutations.

protein interactions between an extracellular loop (*Liu et al., 2019*; *Figure 3A,B*). Notably, this loop is poorly conserved in KCCs (*Figure 3E*). In KCC4, the loop is incompatible with forming a dimer interface without substantial rearrangement due to steric clashes (*Figure 3B*). A triple mutation designed to disrupt interaction between extracellular loops in KCC4 (S404E, R405E, G407E) has no effect on transport activity (*Figure 3B,F*). This mutation (and those described later) did not substantially alter KCC4 folding or expression (*Figure 3—figure supplement 1*). We conclude dimerization as observed in the KCC1 structure (*Liu et al., 2019*) is not functionally relevant for KCC4.

In NKCC1, dimerization is mediated predominantly through extensive protein-protein interactions in the CTD (*Chew et al., 2019*). These regions appear well conserved in KCCs (*Figure 3E*, *Figure 2—figure supplement 1*). Three pairs of mutations designed to disrupt CTD-CTD interactions in KCC4 (L678R, V680R; L670R, A673R; and S780R, L783R) resulted in a similar, but incomplete, reduction in KCC4 activity (by an average of 36, 41, and 44%, respectively) (*Figure 3C,D,F*). This reduction is comparable to that observed in a truncated KCC4 construct missing the entire C-terminal region (KCC4ΔC, which includes amino acids 1–658) (*Figure 3F*). These results suggest that monomeric KCC4 is active and that dimerization through the CTDs in a manner analogous to NKCC1 increases transport activity.

## Transporter conformation

KCC4 adopts an inward-open conformation. The outer surface of the transporter is sealed from the extracellular solution, while a continuous cavity extends from the center of the transmembrane region to the cytoplasmic side (*Figure 4*). The transmembrane region consists of twelve helices (TM1-TM12) with TM1-TM5 related to TM6-TM10 through an inverted repeat. TM2 and TM11 in KCC4 are linked by a membrane buried disulfide bond between amino acids C163 (TM2) and C626 (TM11) conserved between KCCs, but not other CCCs (*Figure 2—figure supplement 1*).

A prominent feature of KCC4 is a large extracellular domain, unique among proteins of known structure, extending ~35 Å above the membrane. It is formed by EL3 (the long

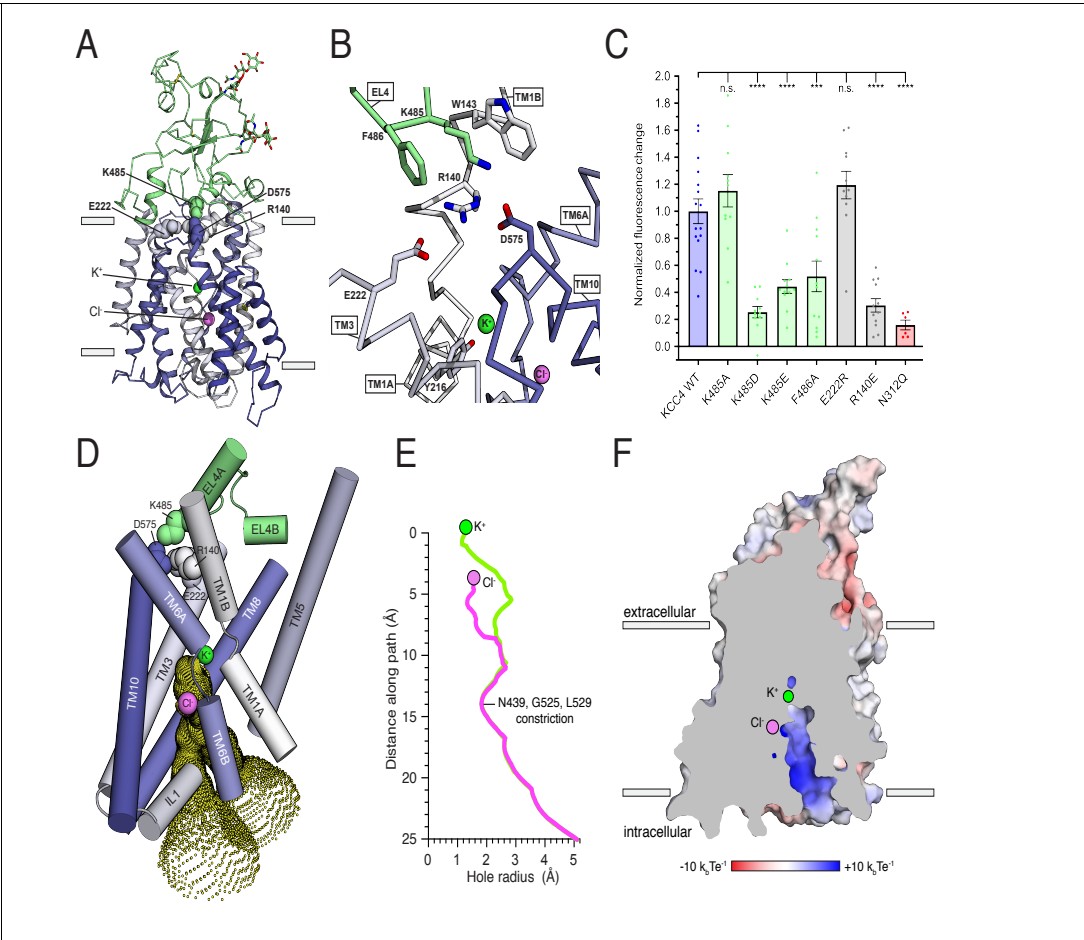

**Figure 4.** Inward-open conformation of KCC4. (**A**) Cartoon representation of KCC4 colored with extracellular region green and transmembrane region colored in a gradient from white to blue from N- to C-terminus. Ions and residues forming an extracellular gate are shown as spheres. (**B**) Close-up view of extracellular gate. Residues forming the interaction network are shown as sticks. (**C**) Normalized activity of KCC4 mutations. Wild-type KCC4 1 ± 0.09 (n = 16); K485A 1.15 ± 0.12 (n = 11); K485D 0.25 ± 0.04 (n = 11); K485E 0.44 ± 0.05 (n = 12); F486A 0.52 ± 0.11 (n = 12); E222R 1.19 ± 0.10 (n = 11); R140E 0.30 ± 0.05 (n = 12); N312Q 0.16 ± 0.04 (n = 6); mean ± SEM, one-way Anova (***p<0.001, ****p<0.0001). (**D**) View of the open pathway to the intracellular ion binding sites. Helices surrounding the ion binding sites are shown as cylinders. Yellow dots demarcate the surface of a bifurcated tunnel that connects the ion binding sites to the cytoplasmic solution. (**E**) Radius of the ion access tunnel as a function of distance along the path for Cl⁻ (pink) and K⁺ (green). (**F**) Electrostatic surface representation of KCC4 sliced to show one leg of the cytoplasmic access tunnel.

extracellular loop; *Hartmann et al., 2010*) and EL4, which pack together and cover ~2/3 of the transporter outer surface (*Figures 2C–D* and *4A*). Sequence comparison suggests it is conserved in all KCCs and not found in other CCCs (*Figure 2—figure supplement 1*). The structure consists of a short three-stranded antiparallel beta sheet (EL4 S1-3), five short helices (EL3 HA-C and EL4 HA,B), and regions without regular secondary structure. It is stabilized by two disulfide bonds (C308-C323 and C343-C352) (*Hartmann et al., 2010*) and decorated with N-linked glycosylation sites (*Marcoux et al., 2017*; *Weng et al., 2013*) conserved among KCCs. Non-protein density consistent with glycosylation is present at four previously identified sites (N312, N331, N344, and N360) (*Weng et al., 2013*) and we model partial carbohydrate chains at the two stronger sites (N312 and N360). Notably, the carbohydrate chain at N312 projects from the EH3 S1-S2 loop underneath an extended segment that leads to TM6A. This arrangement may stabilize the extracellular domain and couple it to movements in TM6A, which moves between functional states in other APC transporters (*Shi, 2013*; *Yamashita et al., 2005*; *Krishnamurthy and Gouaux, 2012*). Indeed, mutation of this site (N312Q) to prevent glycosylation severely reduces KCC4 activity (by an average of 84%, *Figure 4C*). These results provide a structural explanation for functional defects associated with non-glycosylated mutants of KCC4 (*Weng et al., 2013*).

The position of the extracellular domain suggests its involvement in conformational changes during the KCC transport cycle. A segment of the extracellular domain close to the membrane forms a constriction that seals the internal vestibule from the extracellular solution. This is likely the extracellular gate based on comparison to other APC transporters (*Figure 4A,B*; *Shi, 2013*). In KCC4, residues in EL4, TM1, TM3, and TM10 form an electrostatic and hydrophobic interaction network that seals the gate (*Figure 4A,B*). R140 on TM1B extends towards the extracellular solution to interact with D575 on TM10 and E222 on TM3. The extracellular domain is positioned immediately above through an interaction between K485 on EL4 and D575. The outer portion of TM1B contributes W143 which, together with F486, surrounds K485 as it projects towards TM10. This is reminiscent of the extracellular gate in LeuT formed by an electrostatic interaction between TM1 and TM10 (R30 and D404) and capped by EL4 through an interaction with TM10 (D401 and A319) (*Krishnamurthy and Gouaux, 2012*).

We generated mutations at sites in this interaction network to assess its importance for KCC4 function. Disruption of the interaction from the extracellular domain (K485D, K485E, and F486A) or the TM region (R140E) significantly reduced KCC4 transport activity (*Figure 4C*). The functional effects of these mutations are likely due to specific disruption of the interaction between the TM region and extracellular domain because a more subtle change (K485A) or mutation of a nearby, but less conformationally restricted, residue (E222R) had no effect on activity (*Figure 4C*). By analogy to LeuT and other APC transporters, these data suggest opening of the KCC4 extracellular gate likely requires 'unzipping' of the electrostatic network and rotation of EL4 and the extracellular domain away from the surface of the TM region (*Krishnamurthy and Gouaux, 2012*; *Penmatsa and Gouaux, 2014*).

On the intracellular side of KCC4, a hydrophilic cavity is formed by TM1, TM3, TM6, and TM8 that exposes the inside of the transporter to the cytoplasm (*Figure 3C–E*). At the top of this cavity are Cl⁻ and K⁺ binding sites. The cavity forms a bifurcated pathway for ion access to these sites, splitting into two routes approximately halfway through the tunnel due to the position of side chains of N439, R440, and R528. Both sides are open to an essentially equivalent degree (*Figure 4D*). The only constriction outside of the local area surrounding the ions is formed at the position of N439 (from TM6B), G525, and L529 (from TM8) where the cavity narrows to ~3.6 Å in diameter, still sufficiently large for passage of K⁺ and Cl⁻ ions. Within ~3 Å of each ion, the cavity narrows such that it would require at least partial ion dehydration.

The cavity surface is markedly electropositive (*Figure 4E*). From the intracellular solution up to the position of the Cl⁻, charged and polar side chains (from R440, R528, R535, N131, N274, N439, and N521), backbone amides (from IL1), and a helical dipole (from TM6B) contribute electropositive character. Since intracellular Cl⁻ ions are typically present at lower concentrations than K⁺ ions, this may serve to favor accumulation of the less abundant substrate near its binding site within transporter. Above the Cl⁻ site and around the K⁺ site, the accessible surface becomes electronegative and would favor cation binding. The extracellular surface of the transporter outside of the sealed gate is markedly electronegative. How this relates to mechanisms for ion binding and release in outward-open states awaits additional structural information.

## Ion binding sites

The central discontinuities in TM1 and TM6 result in protein backbone carbonyls and amides not involved in regular hydrogen bonding that are utilized in other APC transporters for substrate binding (*Shi, 2013*). Around this region, we observe two prominent non-protein density features (*Figure 5A,D*). Based on structural, functional, and comparative analyses described below, we model these sites as bound K⁺ and Cl⁻ ions.

The stronger of the two densities between TM1, TM6, and TM3 is modeled as a K⁺ ion (*Figure 5A*). It is surrounded by electronegative groups contributed by backbone carbonyls (N131 and I132 in TM1 and P429 and T432 in TM6) and a tyrosine hydroxyl from Y216 in TM3. The distances between electronegative groups and the ion are consistent with K⁺ binding (2.8–2.9 Å). The electronegative helix dipoles created by TM1A and TM6A may additionally contribute to a favorable electrostatic environment for cation binding. The coordinating tyrosine is conserved in all CCC family members that transport K⁺. In NCC, the position corresponding to Y216 is substituted by a histidine, which likely explains its K⁺-independence (*Figure 2—figure supplement 1*).

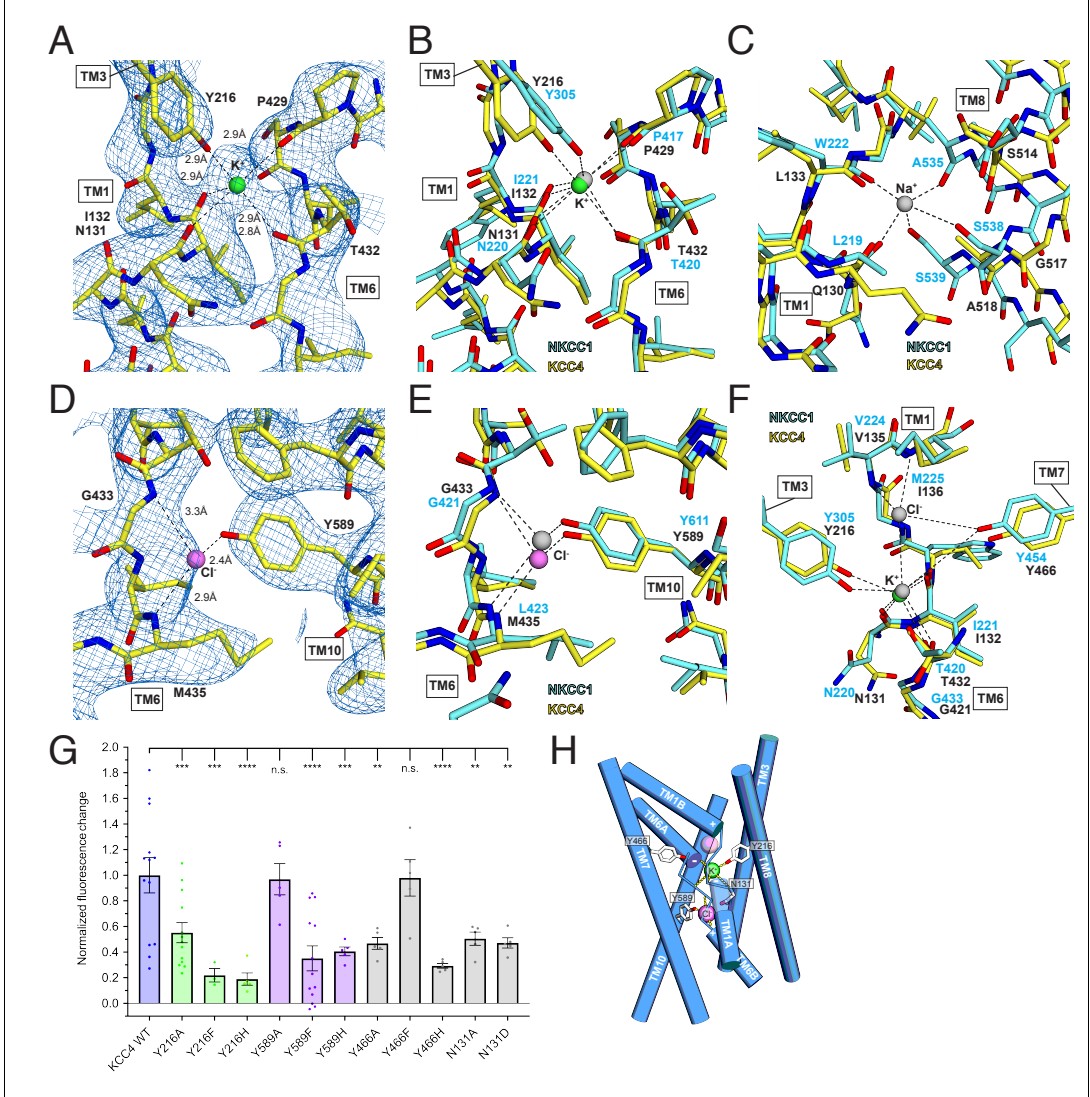

**Figure 5.** Ion binding sites. (**A**) K$^+$ binding site. The cryo-EM map is shown as blue mesh and KCC4 is colored with carbon yellow, oxygen red, nitrogen blue, and K$^+$ green. K$^+$-coordination environment is indicated with dashed lines. (**B**) Superposition of K$^+$ binding sites in KCC4 (depicted as in (**A**)) and *Danio rerio* NKCC1 (PDB 6NPL) colored with carbons cyan and K$^+$ gray. (**C**) Superposition of the Na$^+$ (gray) binding site in NKCC1 and analogous region in KCC4. The position of the Na$^+$ ion is inferred from SiaT (PDB:5NVA). (**D**) Cl$^-$ (pink) binding site and (**E**) superposition of Cl$^-$ binding site in KCC4 with analogous lower site in NKCC1. (**F**) Superposition of second upper Cl$^-$ binding site in NKCC1 with analogous site in KCC4. (**G**) Normalized activity of KCC4 mutations. Wild-type KCC4 $1 \pm 0.14$ (n = 13); Y216A $0.55 \pm 0.08$ (n = 13); Y216F $0.22 \pm 0.05$ (n = 3); Y216H $0.19 \pm 0.05$ (n = 5); Y589A $0.97 \pm 0.12$ (n = 5); Y589F $0.35 \pm 0.10$ (n = 13); Y589H $0.41 \pm 0.03$ (n = 5); Y466A $0.47 \pm 0.05$ (n = 5); Y466F $0.98 \pm 0.14$ (n = 5); Y466H $0.29 \pm 0.02$ (n = 5); N131A $0.50 \pm 0.05$ (n = 5); N131D $0.47 \pm 0.04$ (n = 5); all mean $\pm$ SEM, one-way Anova (**$p<0.01$, ***$p<0.001$ ****$p<0.0001$). (**H**) Model for ion binding and transport stoichiometry in CCC transporters. Helices are shown as cylinders with ion coordination in KCC4 shown as dashed lines to green K$^+$ and pink Cl$^-$. Helix dipoles in discontinuous helices TM1 and TM6 are indicated. A second upper Cl$^-$ site in NKCC1 and KCC1 not observed in the current KCC4 structure is shown as a transparent pink sphere.

The second site, between TM6 and TM10, is modeled as a Cl$^-$ ion (*Figure 5C*). It is surrounded by electropositive groups from backbone amides (G433 and I434 in TM6) and a tyrosine hydroxyl from Y589 in TM10. The electropositive helix dipoles created by TM1B and TM6B may additionally stabilize anion binding. The interaction distances and coordination environment are reminiscent of Cl$^-$

sites in CLC transporters (*Dutzler et al., 2003*) and the coordinating tyrosine is conserved across CCCs.

To validate the assignment of the K$^+$ and Cl$^-$ sites and test the importance of coordinating residues in transporter activity, we mutated Y216 and Y589 that contribute to the binding sites to A, F, and H and assessed transporter activity. Mutations at both sites resulted in a marked loss in transport activity in the Tl$^+$-flux assay: K$^+$-coordinating Y216 mutations Y216A, Y216F, Y216H and Cl$^-$-coordinating Y589 mutations Y589F and Y589H reduced activity (by an average of 45, 78, 81, 65% and 59%, respectively) (*Figure 5G*). Altering the electrostatic character around the Cl$^-$ site with mutations N131A and N131D resulted in comparable transport defects. We conclude the observed K$^+$ and Cl$^-$ sites are critical for KCC4 activity.

How does substrate binding in the 1:1 K$^+$:Cl$^-$ cotransporter KCC4 differ from the 1:1:2 Na$^+$: K$^+$:Cl$^-$ cotransporter NKCC1 (*Warmuth et al., 2009*; *Wahlgren et al., 2018*)? The KCC4 K$^+$ and Cl$^-$ sites correspond closely to sites for the same ions in NKCC1 (*Figure 5B,E*). However, the proposed Na$^+$ site in NKCC1 is dramatically reorganized in KCC4 (*Figure 5C*). In KCC4, TM8 is rotated farther away from TM1 and two consecutive Na$^+$-coordinating serines in NKCC1 (conserved in all Na$^+$-transporting CCCs) are substituted by glycine and alanine in KCC4 (and in KCC1-3) (*Figure 2—figure supplement 1*). The consequence is a loss of three of the five Na$^+$-coordinating positions, providing a structural explanation for Na$^+$-independence in KCCs.

A second Cl$^-$ site in NKCCs extracellular to the K$^+$ site is structurally conserved in KCC4 (*Figure 5F*), but we do not observe evidence for ion occupancy at this site in the structure. Interestingly, this site is occupied in a recent structure of KCC1 in addition to the K$^+$ and Cl$^-$ sites observed in KCC4 (*Liu et al., 2019*). Mutagenesis of a nearby tyrosine Y466A and Y466H reduce activity of KCC4, suggesting that the integrity of both sites is important for transport (*Figure 5G*). Understanding how stoichiometric transport of 1 Cl$^-$ with 1 K$^+$ in KCC4 is accomplished will require structures in additional functional states.

## Discussion

Our structural and mutational data are consistent with prior evidence for monomeric KCCs in cells in addition to homodimers and other heterooligomers (*Blaesse et al., 2006*; *Puskarjov et al., 2012*; *Watanabe et al., 2009*). Comparison of residues in KCC4 that correspond to those in the KCC1 homodimerization interface (*Liu et al., 2019*) suggest this interaction is not relevant for KCC4 function. However, comparison to the NKCC1 homodimerization interface (*Chew et al., 2019*) suggest KCC4 could similarly self-interact and mutational data suggests this interaction increases KCC4 activity. A recently posted report of KCC2 and KCC3 structures appears to show a homodimeric interaction through the CTDs very similar to that observed in NKCC1 (*Chi, 2020*). If monomeric and dimeric KCC4 are functionally distinct, regulated oligomerization in cells would be an opportunity to modulate transport activity. In addition to an increase in basal activity, dimerization involving a close juxtaposition of CTDs and transmembrane regions could enable regulation of transporter activity through CTD posttranslational modifications that may not be possible in monomeric KCC4 with a flexibly attached CTD (*Hartmann and Nothwang, 2014*; *Blaesse et al., 2006*; *Puskarjov et al., 2012*; *Watanabe et al., 2009*). In support of the in vivo relevance of this regulatory mode, a monomeric to dimeric transition in KCC2 has been correlated with an increase in transporter activity (*Blaesse et al., 2006*) and is influenced by phosphorylation of the CTD (*Watanabe et al., 2009*), while proteolytic cleavage of the KCC2 CTD correlates with a decrease in activity (*Puskarjov et al., 2012*). The ability of KCC4 to hetero-oligomerize with CCCs and other membrane proteins may be associated with weaker self-interaction compared to other CCCs (*Simard et al., 2007*; *Mahadevan et al., 2014*; *Goutierre et al., 2019*; *Ivakine et al., 2013*).

The structure of KCC4 provides insight into the architecture of KCCs and the mechanistic basis for coupled K$^+$:Cl$^-$ transport. KCC4 is observed in an inward-open conformation that exposes the inside of the transporter to the cytoplasm through a wide and electropositive tunnel that may serve to concentrate the less abundant intracellular substrate Cl$^-$. This conformation is similar to that observed in NKCC1 and KCC1 and presumably represents the lowest energy state for CCCs in symmetrical salt concentrations and the absence of a transmembrane electrical gradient (*Chew et al., 2019*; *Chi, 2020*). We identify K$^+$ and Cl$^-$ ions around central discontinuities in TM1 and TM6. A Na$^+$

site in the 1:1:2 $Na^+$:$K^+$:$Cl^-$ cotransporter NKCC1 is reorganized in KCC4 due to TM-TM displacement and loss of specific coordinating side chains, explaining $Na^+$-independence in KCCs.

# Materials and methods

## Key resources table

| Reagent type (species) or resource | Designation | Source or reference | Identifiers | Additional information |
|---|---|---|---|---|
| Gene (*Mus musculus*) | KCC4 | Synthesized (Thermo Fisher) | Uniprot Q80WG5 | Codon-optimized for *Spodoptera frugiperda* |
| Gene (*Mus musculus*) | KCC1 | Synthesized (Thermo Fisher) | Uniprot Q9QY75 | Codon-optimized for *Spodoptera frugiperda* |
| Recombinant DNA reagent | pACEBAC1 | Geneva Biotech | pACEBac1 | Modified as described in Materials and methods |
| Cell Line (*Spodoptera frugiperda*) | Sf9 | Expression Systems | Catalog Number: 94–001F | |
| Cell Line (*H. sapiens*) | HEK293T | ATCC | Catalog Number: CRL-3216 | |
| Cell Line (*H. sapiens*) | HEK293T GnTI- | ATCC | Catalog Number: CRL-3022 | |
| Peptide, recombinant protein | MSP1D1 | Prepared as described in doi: 10.1016/S0076-6879(09)64011–8 | | His-tag cleaved |
| Chemical compound | DDM | Anatrace | Part Number: D310S | |
| Chemical compound | CHS | Anatrace | Part Number: CH210 | |
| Chemical compound | Digitonin | EMD Chemicals | CAS 11024-24-1 | |
| Chemical compound | GDN | Anatrace | Part Number: GDN101 | |
| Chemical compound | 18:1 (Δ9-Cis) PE (DOPE) lipid | Avanti Polar Lipids | SKU: 850725C | |
| Chemical compound | 16:0-18:1 PS (POPS) lipid | Avanti Polar Lipids | SKU: 840034C | |
| Chemical compound | 16:0-18:1 PC (POPC) lipid | Avanti Polar Lipids | SKU: 850457C | |
| Chemical compound | FluxOR Red Potassium Ion Channel Assay | ThermoFisher Scientific | Catalog Number: F20019 | |
| Software, algorithm | RELION | DOI: 10.7554/eLife.42166 | Relion 3.0.7 | |
| Software, algorithm | Cryosparc | DOI: 10.1038/nmeth.4169 | Cryosparc2 | |
| Software, algorithm | Ctffind | doi: 10.1016/j.jsb.2015.08.008 | Ctffind 4.1.13 | |
| Software, algorithm | PyEM | doi: 10.5281/zenodo.3576630. | PyEM | https://zenodo.org/record/3576630#.XmptWJNKjUl |
| Software, algorithm | UCSF Chimera | UCSF | RRID:SCR_004097 | http://plato.cgl.ucsf.edu/chimera/ |
| Software, algorithm | COOT | | RRID:SCR_014222 | http://www2.mrc-lmb.cam.ac.uk/personal/pemsley/coot/ |
| Software, algorithm | Phenix | | RRID:SCR_014224 | https://www.phenix-online.org/ |

*Continued on next page*

*Continued*

| Reagent type (species) or resource | Designation | Source or reference | Identifiers | Additional information |
|---|---|---|---|---|
| Software, algorithm | PyMOL | PyMOL Molecular Graphics System, Schrodinger LLC | RRID:SCR_000305 | https://www.pymol.org/ |

## Cloning and protein expression

Cloning, expression, and purification were performed similarly to that described for LRRC8A (*Kern et al., 2019*). The sequence for KCC4 and KCC1 from *Mus musculus* were codon optimized for *Spodoptera frugiperda* and synthesized (Thermo Fisher, Cambridge, MA). Sequences were cloned into a custom vector based on the pACEBAC1 backbone (MultiBac; Geneva Biotech, Geneva, Switzerland) with an added C-terminal PreScission protease (PPX) cleavage site, linker sequence, superfolder GFP (sfGFP), and 7xHis tag, generating a construct for expression of mmKCC4-SNS-LEVLFQGP-SRGGSGAAAGSGSGS-sfGFP-GSS-7xHis. Mutations were introduced using standard PCR techniques with partially overlapping primers. MultiBac cells were used to generate bacmids according to manufacturer's instructions. *Spodoptera frugiperda* (Sf9) cells were cultured in ESF 921 medium (Expression Systems, Davis, CA) and P1 virus was generated from cells transfected with Escort IV Transfection Reagent (Sigma, Carlsbad, CA) according to manufacturer's instructions. P2 virus was generated by infecting cells at $2 \times 10^6$ cells/mL with P1 virus at a MOI ~ 0.1. Infection monitored by fluorescence of sfGFP-tagged protein and P2 virus was harvested at 72 hr post infection. P3 virus was generated in a similar manner to expand the viral stock. The P3 viral stock was then used to infect 1 L of Sf9 cells at $4 \times 10^6$ cells/mL at a MOI ~ 2–5. At 60 hr post-infection, cells were harvested by centrifugation at 2500 x g and frozen at $-80°C$.

## Transporter assay

The FluxOR-Red Potassium Ion Channel Assay (Thermo Fisher Scientific) was adapted for transport assays in Sf9 insect cells by adjusting the osmolarity of all buffers to 380 mOsm (by addition of sodium methylsulfonate). Cells were infected at a density of $1.5 \times 10^6$ cells/ml and grown in suspension for 60–72 hr for robust KCC4-GFP expression. 100 uL of cells at $1 \times 10^6$ cells/ml were plated and allowed to adhere for 1 hr before the assay. For experiments in *Figures 3–5*, 100 uL of cells at $1 \times 10^6$ cells/ml were plated and allowed to adhere for 1 hr before viral infection. Cells were grown for 60–72 hr for robust transporter expression prior to assay. Growth media was replaced with 1X Loading Buffer and incubated at 27°C away from light for 1 hr. The FluxOR Red reagent is a non-fluorescent indicator dye which is loaded into cells as a membrane-permeable acetoxymethyl (AM)-ester. The non-fluorescent AM ester of the FluxOR Red reagent is cleaved by endogenous esterases into a fluorogenic $Tl^+$-sensitive indicator. 1X Loading Buffer was subsequently removed and replaced with Dye-free Assay Buffer and FluxOR Background Suppressor. The assay was performed in 96-well, black-walled, clear-bottom plates (Costar). For data in *Figure 1*, fluorescence was measured on a Perkin-Elmer Envision Multilabel Plate Reader using bottom read fluorescence and a BODIPY TMR FP filter set (excitation 531 nm and 25 nm bandwidth, emission 595 nm and 60 nm bandwidth). For data in *Figures 3–5*, Fluorescence was measured on a Molecular Devices SpectraMax M3 using bottom read fluorescence and an excitation 560 nm with auto cutoff at 590 nm and emission 590 nm. The recordings were baseline corrected by subtracting the average fluorescence from 180 s prior to the addition of Basal Potassium Stimulus buffer and time zero is defined as the first data point recorded after the addition of stimulus. Global fits of all data to a one phase association model $Y= (Plateau)*(1-e^{(-x/\tau)})$ are displayed with 95% confidence interval bands (*Figure 1B*). Alternatively, the final 50 counts were averaged as a measure of final fluorescence increase and normalized to wild-type KCC4 fluorescence increase from experiments performed on the same day. In some experiments, average GFP fluorescence was measured between FluxOR fluorescence measurements and used to normalize FluxOR fluorescence values to account for differences in expression level. This normalization did not change results in a significant way and so was not performed for the final data presented.

## Protein purification

Cells from 1 L of culture (~7–12.5 mL of cell pellet) were thawed in 100 mL of Lysis Buffer (50 mM Tris, 150 mM KCl, 1 mM EDTA, pH 8.0). Protease inhibitors were added to the lysis buffer immediately before use (final concentrations: E64 (1 µM), Pepstatin A (1 µg/mL), Soy Trypsin Inhibitor (10 µg/mL), Benzimidine (1 mM), Aprotinin (1 µg/mL), Leupeptin (1 µg/mL), and PMSF (1 mM)). Benzonase (5 µl) was added after cells thawed. Cells were then lysed by sonication and centrifuged at 150,000 x g for 45 min. The supernatant was discarded, and residual nucleic acid was removed from the top of the membrane pellet by rinsing with DPBS. A 10%/2% and 10%/1% w/v solution of DDM/CHS was clarified by bath sonication in 200 mM Tris pH 8.0 and subsequently added to buffers at the indicated final concentrations. Membrane pellets were transferred to a glass dounce homogenizer containing Extraction Buffer (50 mM Tris, 150 mM KCl, 1 mM EDTA, 1% w/v n-Dodecyl-β-D-Maltopyranoside (DDM, Anatrace, Maumee, OH), 0.2% w/v Cholesterol Hemisuccinate Tris Salt (CHS, Anatrace), pH 8.0). Membrane pellets were homogenized in Extraction Buffer and this mixture (100 mL final volume) was gently stirred at 4°C for 1 hr. The extraction mixture was centrifuged at 33,000 x g for 45 min. The supernatant, containing solubilized KCC4-sfGFP, was bound to 5 mL of Sepharose resin coupled to anti-GFP nanobody for 1 hr at 4°C. The resin was collected in a column and washed with 20 mL of Buffer 1 (20 mM Tris, 150 mM KCl, 1 mM EDTA, 0.025% DDM, 0.005% CHS, pH 8.0), 50 mL of Buffer 2 (20 mM Tris, 500 mM KCl, 1 mM EDTA, 0.025% DDM, 0.005% CHS, pH 8.0), and 20 mL of Buffer 1. Washed resin was resuspended in 6 mL of Buffer 1 with 0.5 mg of PPX and rocked gently in the capped column overnight. Cleaved KCC4 protein was eluted with an additional 25 mL of Buffer 1. The eluted pool was concentrated to ~500 µl with an Amicon Ultra spin concentrator 100 kDa cutoff (MilliporeSigma, USA) and subjected to size exclusion chromatography using a Superose 6 Increase column (GE Healthcare, Chicago, IL) run in Buffer 3 (20 mM Tris pH 8.0, 150 mM KCl, 1 mM EDTA, 0.025% DDM, 0.0025% CHS) on a NGC system (Bio-Rad, Hercules, CA). Peak fractions containing KCC4 transporter were collected and concentrated.

## Fluorescence Size Exclusion Chromatography (FSEC)

Sf9 cells were plated at $1 \times 10^6$ cells/ml into six well plates and allowed to adhere for 1 hr prior to viral infection at a ratio of 1:30 (v/v). Cells were harvested after 60–72 hr, pelleted by centrifugation, and frozen. Transfected HEK 293T GNTI- cells were prepared using Lipofectamine 2000 according to manufacturer's instructions. Media was switched 18 hr post-transfection to fresh media with 10 mM sodium butyrate. Cells were incubated for 19 hr longer at 30°C, harvested, pelleted by centrifugation, and frozen.

Frozen samples containing ~8 million infected Sf9 cells and ~1 million transfected HEK 293T GNTI- cells were thawed, extracted for 1 hr at 4°C, and pelleted at 21,000 x g at 4°C for 1 hr. Supernatant was run on a Superose 6 Increase column with fluorescence detection for GFP. For mutant comparisons (*Figure 3—figure supplement 1*), extraction buffer was (50 mM Tris pH 8, 150 mM KCl, 1 mM EDTA, all protease inhibitors used for protein purification, 1% DDM, 0.2% CHS) and running buffer was (20 mM Tris pH8, 150 mM KCl, 1 mM EDTA, 0.025% DDM, 0.0025% CHS). The same buffers were used for the 'DDM/CHS' conditions in *Figure 2—figure supplement 8A,B,C* and in *Figure 2—figure supplement 8D,E* except that running buffer in *Figure 2—figure supplement 8E* contained (0.025% DDM, 0.005% CHS). The 'GDN' condition used in *Figure 2—figure supplement 8A,B,C* corresponds to conditions used for KCC1 structure determination (*Liu et al., 2019*). Extraction buffer was (20 mM Tris pH8, 150 mM KCl, all protease inhibitors used for protein purification, 2% DDM, 0.2% CHS). Running buffer was (20 mM Tris pH8, 150 mM KCl, 0.06% GDN). The 'Digitonin' condition used in *Figure 2—figure supplement 8A,B,C* corresponds to conditions used for NKCC1 structure determination (*Chew et al., 2019*). Extraction buffer was (50 mM Tris pH 8, 150 mM KCl, 1 mM EDTA, all protease inhibitors used for protein purification, 1% LMNG, 0.01% CHS) and running buffer was (20 mM Tris pH8, 150 mM KCl, 1 mM EDTA, 0.06% Digitonin).

## Cross linking and mass spectrometry

Fractions corresponding to peaks 1 and 2 from size exclusion chromatography were separately pooled and concentrated to 0.5 mg/mL. Crosslinking was performed by adding 1 uL of glutaraldehyde from 10X stock solutions in water to 10 uL of KCC4 to achieve final glutaraldehyde concentrations of 0.02, 0.01, 0.005, 0.0025, and 0%. Samples were incubated for 30 min prior to quenching by

addition of 1 uL 1M Tris-HCl and analysis by SDS-PAGE on 4–12% Tris-glycine gel (BioRad, USA). Deglycosylated samples were pretreated with 1:10 vol purified PNGase at 1 mg/mL (Addgene 114274) for 1 hr at 4°C prior to the addition of glutaraldehyde.

For mass spectrometry, the band corresponding to purified KCC4 was excised from a 4–12% Tris-glycine gel, digested with trypsin in situ, and the resulting peptides extracted and concentrated. Mass spectrometry was performed by the Vincent J. Coates Proteomics/Mass Spectrometry Laboratory at UC Berkeley. A nano LC column was packed in a 100 μm inner diameter glass capillary with an emitter tip. The column consisted of 10 cm of Polaris c18 5 μm packing material (Varian), followed by 4 cm of Partisphere 5 SCX (Whatman). The column was loaded by use of a pressure bomb and washed extensively with buffer A (5% acetonitrile/0.02% heptaflurobutyric acid (HBFA)). The column was then directly coupled to an electrospray ionization source mounted on a Thermo-Fisher LTQ XL linear ion trap mass spectrometer. An Agilent 1200 HPLC equipped with a split line so as to deliver a flow rate of 300 nl/min was used for chromatography. Peptides were eluted using a 4-step MudPIT procedure (*Washburn et al., 2001*). Buffer A was 5% acetonitrile/0.02% heptaflurobutyric acid (HBFA); buffer B was 80% acetonitrile/0.02% HBFA. Buffer C was 250 mM ammonium acetate/5% acetonitrile/0.02% HBFA; buffer D was same as buffer C, but with 500 mM ammonium acetate.

Protein identification was done with Integrated Proteomics Pipeline (IP2, Integrated Proteomics Applications, Inc San Diego, CA) using ProLuCID/Sequest, DTASelect2 and Census (*Xu et al., 2015*; *Tabb et al., 2002*; *Park et al., 2008*). Tandem mass spectra were extracted into ms1 and ms2 files from raw files using RawExtractor (*McDonald et al., 2004*). Data was searched against a *Spodoptera frugiperda* protein database with the purified mouse KCC4 sequence added, supplemented with sequences of common contaminants, and concatenated to form a decoy database (*Peng et al., 2003*). LTQ data was searched with 3000.0 milli-amu precursor tolerance and the fragment ions were restricted to a 600.0 ppm tolerance. All searches were parallelized and searched on the VJC proteomics cluster. Search space included all half tryptic peptide candidates with no missed cleavage restrictions. Carbamidomethylation (+57.02146) of cysteine was considered a static modification. In order to identify authentic termini, we required one tryptic terminus for each peptide identification. The ProLuCID search results were assembled and filtered using the DTASelect program with a peptide false discovery rate (FDR) of 0.001 for single peptides and a peptide FDR of 0.005 for additional peptides for the same protein. Under such filtering conditions, the estimated false discovery rate was less than 1%.

## Nanodisc reconstitution

Freshly purified and concentrated KCC4 in Buffer three was reconstituted into MSP1D1 nanodiscs with a mixture of lipids (DOPE:POPS:POPC at 2:1:1 molar ratio, Avanti, Alabaster, Alabama) at a final molar ratio of KCC4:MSP1D1:lipids of 0.2:1:50. Lipids in chloroform were prepared by mixing, drying under argon, washing with pentane, drying under argon, and placing under vacuum overnight. The dried lipid mixture was rehydrated in Buffer 4 (20 mM Tris, 150 mM KCl, 1 mM EDTA pH 8.0) and clarified by bath sonication. DDM was added to a final concentration of 8 mM and the detergent solubilized lipids were sonicated until clear. Lipids, Buffer 4 containing 8 mM DDM, and KCC4 protein were mixed and incubated at 4°C for 30 min before addition of purified MSP1D1. After addition of MSP1D1, the nanodisc formation solution was 47.5 μM KCC4, 104 μM MSP1D1, 13 mM DOPE:POPS:POPC, and 4 mM DDM in Buffer 4 (final concentrations). After mixing at 4°C for 30 mins, 60 mg of Biobeads SM2 (Bio-Rad, USA) (prepared by sequential washing in methanol, water, and Buffer four and weighed damp following bulk liquid removal) were added and the mixture was rotated at 4°C overnight (~12 hr). Nanodisc-containing supernatant was collected and spun for 10 min at 21,000 x g before loading onto a Superose 6 Increase column in Buffer 4. Peak fractions corresponding to KCC4-MSP1D1 were collected and spin concentrated using a 100 kDa cutoff for grid preparation.

## Grid preparation

The KCC4-MSP1D1 nanodisc sample was concentrated to ~1 mg/mL and centrifuged at 21,000 x g for 10 min at 4°C prior to grid preparation. A 3 uL drop of protein was applied to a freshly glow discharged Holey Carbon, 400 mesh R 1.2/1.3 gold grid (Quantifoil, Großlöbichau, Germany). A Vitrobot Mark IV (FEI/Thermo Scientific, USA) was utilized for plunge freezing in liquid ethane with the

following settings: 4°C, 100% humidity, one blot force, 3 s blot time, 5 s wait time. The KCC4 detergent sample was frozen at 4.5 mg/mL and centrifuged at 21,000 x g for 10 min at 4°C prior to grid preparation. A 3 μL drop of protein was applied to a freshly glow discharged Holey Carbon, 400 mesh R 1.2/1.3 gold grid. A Vitrobot Mark IV (FEI/Thermo Scientific, USA) was utilized for plunge freezing in liquid ethane with the following settings: 4°C, 100% humidity, one blot force, 4 s blot time, 1 s wait time. Grids were clipped in autoloader cartridges for data collection.

## Data collection

KCC4-MSP1D1 grids were transferred to a Talos Arctica cryo-electron microscope (FEI/Thermo Scientific, USA) operated at an acceleration voltage of 200 kV. Images were recorded in an automated fashion with SerialEM (*Mastronarde, 2005*) using image shift with a target defocus range of −0.7 ~ −2.2 μm over 5 s as 50 subframes with a K3 direct electron detector (Gatan, USA) in super-resolution mode with a super-resolution pixel size of 0.5685 Å. The electron dose was 9.333 e⁻ / Å (*Marcoux et al., 2017*)/s (0.9333 e⁻ / Å2/frame) at the detector level and total accumulated dose was 46.665 e-/Å2. KCC4-detergent grids were transferred to a Titan Krios cryo-electron microscope (FEI/Thermo Scientific, USA) operated at an acceleration voltage of 300 kV. Images were recorded in an automated fashion with SerialEM (*Mastronarde, 2005*) with a target defocus range of −0.7 to −2.2 μm over 9.6 s as 48 subframes with a K2 direct electron detector (Gatan, USA) in super-resolution mode with a super-resolution pixel size of 0.5746 Å. The electron dose was 6.092 e⁻ / Å (*Marcoux et al., 2017*)/s (1.2184 e⁻ / Å (*Marcoux et al., 2017*)/frame) at the detector level and total accumulated dose was 58.4832 e⁻/Å (*Marcoux et al., 2017*). See also *Table 1* for data collection statistics.

## Data processing

The processing pipeline is shown in *Figure 2—figure supplement 4A–C*. We used Cryosparc2 (*Punjani et al., 2017*) for initial model generation and refinement until reconstructions reached 4–5 Å resolution. Bayesian polishing and nanodisc subtraction in Relion 3.0.7 (*Zivanov et al., 2019*; *Zivanov et al., 2018*) were used to achieve highest resolution reconstructions. While the contribution of disordered or flexible N- and C-terminal regions to alignments is unknown, the remaining 55 kDa asymmetric membrane protein is among the smallest in terms of resolved mass resolved by cryo-EM to date.

A total of 1572 movie stacks were collected, motion-corrected and binned to 1.137 Å/pixel using MotionCor2 (*Zheng et al., 2017*), and CTF-corrected using Ctffind 4.1.13 (*Rohou and Grigorieff, 2015*; *Figure 2—figure supplement 4A*). Micrographs with a Ctffind reported resolution estimate worse than 5 Å were discarded. A small number of particles (~1000) were picked manually and subjected to two-dimensional classification to generate references for autopicking in Relion. 1,826,000 particles were autopicked and extracted at 2.274 Å/pixel (2x binned) for initial cleanup. Non-particle picks and apparent junk particles were removed by several rounds of two-dimensional class averaging. The remaining 887,132 particles were extracted at 1.137 Å/pixel and imported into Cryosparc. An additional round of 2D classification generated a particle set of 491,111. These particles were the input of an ab initio reconstruction (non-default values: four classes, 0.1 class similarity, 4 Å max resolution, per-image optimal scales). 2D classification of particles (160,868) that contributed to the most featured volume resulted in a set of 125,593 particles which were the input of an ab initio reconstruction. Alignments were iteratively improved using non-uniform (NU) refinement (0.89 window inner radius, 120 voxel box size, 10 extra final passes, 10 Å low-pass filter, 0.01 batch epsilon, minimize over per-particle scale, 1–4 Å dynamic mask near, 3–8 Å dynamic mask far, 6–10 Å dynamic mask start resolution, 4–6 Å local processing start resolution). Two separate NU refinement output volumes were input into a heterogeneous refinement job of the 886,528 particle set (forced hard classification, 10 Å initial resolution, five final full iterations). The more featured class (538,280 particles) was heterogeneously refined (forced hard classification, 10 Å initial resolution, five final full iterations) (*Figure 2—figure supplement 4B*). The particles and volume from one output (354,234 particles) were input into the first of three iterative NU refinements (10 extra final passes for the second and third iteration).

Particle positions and angles from the final cryoSPARC2 refinement job were input into Relion (using csparc2relion.py from the UCSF PyEM [*Asarnow, 2016*]) and 3D refined to generate a 4.18 Å

**Table 1.** Cryo-EM data collection and structure refinement statistics.

| Data collection | |
|---|---|
| Total movie # | 1572 |
| Selected movie # | 1401 |
| Magnification | 36,000x |
| Voltage (kV) | 200 |
| Electron exposure (e⁻/Å²) | 46.665 |
| Frame # | 50 |
| Defocus range (µm) | −0.7 to −2.5 |
| Super resolution pixel size (Å) | 0.5685 |
| Binned pixel size (Å) | 1.137 |
| **Processing** | |
| Initial particle images (no.) | 887,132 |
| Final particle images (no.) | 110,143 |
| **Map resolution** | |
| Masked (Å, FSC = 0.143/ FSC=0.5) | 3.6/4.2 |
| Unmasked (Å, FSC = 0.143/ FSC=0.5) | 3.9/4.4 |
| **Refinement** | |
| Model resolution (Å, FSC = 0.143/ FSC=0.5) | 3.5/3.9 |
| Map-sharpening $B$ factor (Å²) | −150 |
| **Composition** | |
| Number of atoms | 4103 |
| Number of protein residues | 536 |
| Ligands total | 4 |
| K+ | 1 |
| Cl- | 1 |
| NAG-NAG-BMA | 2 |
| **R.m.s. deviations** | |
| Bond lengths (Å) | 0.005 |
| Bond angles (°) | 0.731 |
| **Validation** | |
| MolProbity score | 1.7 |
| Clashscore | 4.59 |
| EMRinger score | 1.74 |
| **Ramachandran plot** | |
| Favored (%) | 92.48 |
| Allowed (%) | 7.52 |
| Disallowed (%) | 0 |
| Rotamer outliers (%) | 0.23 |
| **Mean $B$ factors (Å²)** | |
| Protein | 75.41 |
| Ligand | 106.62 |

map (6 Å low-pass filter, 0.9 degrees initial sampling, 0.9 degrees local searches) (*Figure 2—figure supplement 4C*). A second 3D refinement following Bayesian particle polishing improved the map and reported resolution (4.01 Å) (6 Å low-pass filter, 0.9 degrees initial sampling, 0.9 degrees local searches). CTF refinement with beam tilt group estimation and per-particle defocus was performed,

although subsequent 3D refinement did not markedly improve the map. Particle subtraction was performed to remove the contribution of the nanodisc density from alignments and subsequent 3D refinement markedly improved the map (reported resolution 3.86 Å or 3.72 Å after postprocessing) (6 Å low-pass filter, 0.9 degrees initial sampling, 0.9 degrees local searches). A final improvement in map quality and reported resolution and was obtained by removing poor particles with a 3D classification job (two classes, 10 Å initial low-pass filter, 16 tau fudge, no angular sampling). The final particle set (110,143) was subjected to 3D refinement to generate a final map at 3.72 Å resolution (3.65 Å after postprocessing) (6 Å low-pass filter, 0.9 degrees initial sampling, 0.9 degrees local searches). Particle distribution and local resolution was calculated using Relion (*Figure 2—figure supplement 5A,B*). FSCs reported in *Figure 2—figure supplement 5* were calculated using Phenix.mtriage.

## Modeling, refinement, and structure analysis

The final cryo-EM maps were sharpened using Phenix.autosharpen (*Adams et al., 2010*). The structure was modeled de novo in Coot and refined in real space using Phenix.real_space_refine with Ramachandran and NCS restraints. Validation tools in Phenix, EMRinger (*Barad et al., 2015*), and Molprobity (*Chen et al., 2010*) were used to guide iterative rounds of model adjustment in Coot and refinement in Phenix. Cavity measurements were made with HOLE implemented in Coot (*Emsley et al., 2010*). Electrostatic potential was calculated using APBS-PDB2PQR (*Dolinsky et al., 2004*) Pymol plugin. Figures were prepared using PyMOL, Chimera, ChimeraX, Fiji, Prism, Adobe Photoshop, and Adobe Illustrator software.

## Acknowledgements

We thank members of the Brohawn and Eunyong Park laboratories for feedback and critical reading of the manuscript. We thank Dr. Dan Toso, Dr. Jonathon Remis, and Paul Tobias at the Berkeley Bay Area Cryo-EM facility and UC Berkeley Talos Arctica facility for assistance with microscope setup and data collection. We thank Dr. Lori Kohlstaedt for assistance with mass spectrometry. This work used the Vincent J Proteomics/Mass Spectrometry Laboratory at UC Berkeley, supported in part by NIH S10 Instrumentation Grant S10RR025622. We thank Dr. Mary West and Dr. Pingping He of the High-Throughput Screening Facility (HTSF) at UC Berkeley. This work was performed in part in the HTSF, which provided the Perkin-Elmer Envision Multilabel Plate Reader. SGB is a New York Stem Cell Foundation-Robertson Neuroscience Investigator. This work is supported by a UC Berkeley Chancellor's Fellowship (MSR), and a NIGMS postdoctoral fellowship F32GM128263 (DMK), the New York Stem Cell Foundation (SGB), a NIGMS grant DP2GM123496-01 (SGB), a McKnight Foundation Scholar Award (SGB), and a Klingenstein-Simons Foundation Fellowship Award (SGB).

## Additional information

### Funding

| Funder | Grant reference number | Author |
| --- | --- | --- |
| New York Stem Cell Foundation | NYSCF-R-N145 | Stephen Graf Brohawn |
| National Institute of General Medical Sciences | DP2GM123496 | Stephen Graf Brohawn |
| McKnight Endowment Fund for Neuroscience | 043108 | Stephen Graf Brohawn |
| Klingenstein Third Generation Foundation | | Stephen Graf Brohawn |
| National Institute of General Medical Sciences | F32GM128263 | David M Kern |
| University of California, Berkeley | Chancellor's Fellowship | Michelle S Reid |

The funders had no role in study design, data collection and interpretation, or the decision to submit the work for publication.

## Author contributions
Michelle S Reid, David M Kern, Data curation, Formal analysis, Validation, Investigation, Visualization, Methodology; Stephen Graf Brohawn, Conceptualization, Data curation, Formal analysis, Supervision, Funding acquisition, Validation, Visualization, Project administration

## Author ORCIDs
Michelle S Reid ⓘ https://orcid.org/0000-0002-6947-6053
David M Kern ⓘ https://orcid.org/0000-0001-8529-9045
Stephen Graf Brohawn ⓘ https://orcid.org/0000-0001-6768-3406

## Decision letter and Author response
Decision letter https://doi.org/10.7554/eLife.52505.sa1
Author response https://doi.org/10.7554/eLife.52505.sa2

# Additional files

## Supplementary files
• Transparent reporting form

## Data availability
The final map of KCC4 in MSP1D1 nanodiscs has been deposited to the Electron Microscopy Data Bank under accession code EMD-20807. Atomic coordinates have been deposited in the PDB under ID 6UKN. Original KCC4 in MSP1D1 nanodiscs micrograph movies have been deposited to EMPIAR under deposition EMPIAR-10394.

The following datasets were generated:

| Author(s) | Year | Dataset title | Dataset URL | Database and Identifier |
|---|---|---|---|---|
| Reid MS, Kern DM, Brohawn SG | 2020 | Cryo-EM structure of the potassium-chloride cotransporter KCC4 in lipid nanodiscs | https://www.ebi.ac.uk/pdbe/emdb/empiar/entry/10394 | Electron Microscopy Public Image Archive, 10394 |
| Reid MS, Kern DM, Brohawn SG | 2019 | Cryo-EM structure of the potassium-chloride cotransporter KCC4 in lipid nanodiscs | http://www.ebi.ac.uk/pdbe/entry/emdb/EMD-20807 | Electron Microscopy Data Bank, EMD-20807 |
| Reid MS, Kern DM, Brohawn SG | 2019 | Cryo-EM structure of the potassium-chloride cotransporter KCC4 in lipid nanodiscs | http://www.rcsb.org/structure/6UKN | RCSB Protein Data Bank, 6UKN |

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
