## [Decision Letter]

**Acceptance summary:**

The CCC family of sodium-potassium-chloride cotransporters belong to the SLC12 family of solute carriers, which play a fundamental role in regulating cellular ion homeostasis. Here, Reid et al. report the cryo-EM structure of KCC4, a member of the electroneutral potassium-chloride cotransporter subfamily, from *Mus musculus* in lipid nanodiscs. The structural data reveal differences in the ion-binding sites between KCC and NKCC members, explaining differences in stoichiometry and ion preferences in the larger SLC12 family. Analysis of the extracellular domain in KCC4 builds on previous structures from KCC1, and highlights an intriguing role for glycosylation in regulating transport function. Functional studies demonstrated that KCC4 could function as both monomer and dimer in the cell, with dimerization increasing activity and possibly linked to activation during cell swelling.

**Decision letter after peer review:**

[Editors’ note: the authors submitted for reconsideration following the decision after peer review. What follows is the decision letter after the first round of review.]

Thank you for submitting your work entitled "Cryo-EM structure of the potassium-chloride cotransporter KCC4 in lipid nanodiscs" for consideration by *eLife*. Your article has been reviewed by three peer reviewers, and the evaluation has been overseen by a Reviewing Editor and a Senior Editor. The following individuals involved in review of your submission have agreed to reveal their identity: Jue Chen (Reviewer #1) and Etana Padan (Reviewer #3).

Our decision has been reached after consultation between the reviewers. Based on these discussions and the individual reviews below, we regret to inform you that we cannot consider your work for publication in *eLife* at this stage. We would be happy to consider a resubmission, if you revise your manuscript along the lines suggested below.

Summary:

The manuscript by Reid et al. reports the structure of the mammalian potassium chloride cotransporter KCC4, determined by single-particle cryo-EM in lipid nanodiscs. KCC4 is member of the larger Cation Chloride Cotransporter superfamily, which plays essential roles in regulating ion homeostasis in the cell. The structure comes hot on the heels of other CCC family members, NKCC1 and KCC1, which makes it potentially a valuable addition to the study of these physiologically important antiporters. As it stands, the manuscript does however not present a sufficient advance over recently published work.

Our main concerns were as follows:

1) While all reviewers find the paper interesting and potentially important, they are concerned that this is the third structure of this family of transporters. The other two, which were published earlier this year, are human KCC1 and DrNKCC1. In your structure, the transmembrane domain is nearly identical to the other two, including the ion binding sites. The major difference is that KCC4 is a monomer and the others are dimers. This would be an opportunity to understand dimerization as a new aspect of these transporters. Unfortunately this difference is only addressed superficially in the discussion but without data or analysis. If you can provide evidence that the monomeric form of KCC4 is active, for example by ITC, it would be a valuable addition. If you could show that the ion transport or ion binding kinetics for the monomer and dimer are different, this would add considerable weight to your study.

2) A related concern was that your study provides little in the way of insight into the regulation of the protein. While you propose that monomer-dimer transitions may occur during regulation of the protein in cells, no evidence is provided that this is actually the case.

3) The fact that KCC4 is a monomer is surprising, given that the KCC1 structure, which is closely related, was determined as a dimer using a similar technique. Could the use of 1% DDM be responsible for disrupting the dimer during isolation? Native mass spectrometry might be a good way of analyzing this important aspect of KCC structure and regulation.

4) A further point that would benefit from additional analysis is the role of the extracellular domain. You describe this domain as novel, but without investigating its function, either in cells or in vitro. You describe glycosylation, including what appears to be an interesting link between glycosylation at N312 and TM6A, which forms part of the transport machinery in APC proteins. Yet, the significance of this interaction is not clear. Is it an artifact or an important part of the mechanism?

5) We also wondered about the extracellular gate, where you describe a salt bridge network that may function in a similar way to LeuT (and other APC transporters), but no analysis is provided. It would be good to characterize these interactions, for example through transport assays with variants at these sites and charge-swapped mutants. Such experiments would illuminate the mechanism of KCC4 and help to put the structure into a functional context.

Reviewer #1:

This paper describes the cryo-EM structure of KCC4, a K^+^/Cl^-^ cotransporter from *Mus musculus*. The structural work is well done, two residues in the ion-binding sites were mutated to demonstrate their importance to transport activity.

My main concern is this:

This is the third structure of this family of transporters. The other two, which were published earlier this year, are human KCC1 and DrNKCC1. The new structure in detail is nearly identical to the previous ones, including the ion binding sites. Some of the figures here are also similar. The major difference is that KCC4 is a monomer and the others are dimers. This is an opportunity to understand an aspect of these transporters that until now was unknown. Unfortunately this difference is only addressed superficially in discussion but not with data and analysis. If the authors can provide evidence that the monomeric form of KCC4 is functional, then this work will make a good contribution towards our understanding of the architecture and function of KCCs.

Reviewer #2:

The study by Reid et al. reports the structure of the mammalian potassium chloride cotransporter KCC4, determined using single particle Cryo-EM in lipid nanodiscs. KCC4 is member of the larger Cation Chloride Cotransporter superfamily, members of which play essential roles in regulating ion homeostasis in the cell. This structure comes hot on the heels of other CCC family members, NKCC1 and KCC1, which makes the current structure a valuable addition to the literature on these physiologically important antiporters. The structure appears to be of good quality and is accompanied by cross linking and mass spectrometry data to investigate the oligomeric state and cell based transport assays to understand mechanism. The manuscript is clearly written and the figures are of good quality and appropriate for the description.

My main concern with the current manuscript is that apart from reporting the structure the study provides little else in the way of insight into the mechanism or regulation of the protein. I felt the manuscript was very descriptive in format but without any significant interpretation. The cross linking analysis appears to have provided little in the way of insight in to the oligomeric state for example. While the authors propose that monomer dimer transitions may occur during regulation of the protein in cells, no evidence is provided that this protein is capable of dimerization. This is rather odd, especially given that the KCC1 structure, which is closely related, was determined as a dimer using a similar technique. I wondered whether the use of 1% DDM might be responsible for disrupting the dimer interaction? I also wondered whether native mass spectrometry might be a better way of analyzing this important aspect of KCC regulation.

Another area of the current study that I found frustrating was the lack of analysis on the role of the extracellular domain. For example, the authors describe the novel extracellular domain, but do not investigate its function, either in cells or in vitro. They describe the glycosylation observed, including what appears to be an interesting link between glycosylation at N312 and TM6A, which forms part of the transport machinery in APC proteins. Yet, the study leaves the reader wondering what the significance of this interaction is. Is it an artifact or an important part of the mechanism? I wondered the same when reading the section that describes the extracellular gate. Here the authors describe a set of salt bridge networks that appear to function similar to LeuT (and other APC transporters), but no analysis is provided. I would have liked to see an attempt made to functionally characterize these interactions, either through transport assays with variants at these sites and charge swapped mutants. Such experiments would greatly illuminate the mechanism of KCC4 and help to put the structure in the context of the function. The same could be said of the bifurcated pathway – artifact or interesting mechanistic observation. Here mutations and molecular dynamics could shed more insight.

In relation to the comparison between KCC4 and NKCC4, the latter protein binds to Na^+^ in addition to K^+^ and Cl^-^, I was interested to know what compensatory interactions are made that enables KCC4 to dispense with sodium binding. In the APC transporters that move protons for example, the second Na^+^ site is substituted with a lysine. I felt this part of the analysis was also less in depth that warranted.

Overall the manuscript reports the structure of an important membrane protein that will be of interest to a wide range of researchers. However, the study for me was let down by a lack of functional analysis of the protein, starting with the oligomeric state and leading into analysis of the interesting structural features observed. Analysis of the ion binding sites is useful, but ultimately could be seen as fairly obvious and confirmatory, especially given the number of similar APC structures in the literature and the in-depth biochemical analysis conducted on these proteins.

Reviewer #3:

This manuscript describes the structure of the potassium-chloride cotransporter KCC4 (of *Mus musculus*) in lipid nanodiscs using single particle Cryo-EM. This cotransporter belongs to the CCCs family of which members play important roles in many human physiological functions. Several of these transporters are drug targets because their disruptions are involved in pathological cases. Hence, the KCC4 structure is important both for understanding the structure and function of an important group of mammal's transporters as well as for drug design.

1) KCC4 induced Tl^+^ uptake activity has been demonstrated in intact cells. However, dependence on the symported ion-Cl has only been illustrated. Please, clarify why?

2) Although, the Tl^+^ uptake due to KCC4 activity was nicely determine in intact cells the structure was determined in nanodiscs. Therefore, there is no indication that the revealed structure is the native functional protein structure. Activity of the protein in nanodiscs is missing and this can be accomplished by testing binding of K^+^/Cl^-^. Using ITC is a simple option.

3) Structure based mutants support the proposed binding sites and have raised interesting discussion regarding ion/substrate selectivity in this group of transporters. The authors also suggest that amino acid residues residing in the protein's extracellular domain form an outward gate of the transporter. This interesting suggestion has to be explored by analyzing mutations at least of the residues that are localized outside the membrane part of the transporter.

---

## [Author Response]

[Editors’ note: The authors appealed the original decision. What follows is the authors’ response to the first round of review.]

Our main concerns were as follows:1) While all reviewers find the paper interesting and potentially important, they are concerned that this is the third structure of this family of transporters. The other two, which were published earlier this year, are human KCC1 and DrNKCC1. In your structure, the transmembrane domain is nearly identical to the other two, including the ion binding sites.

While the three CCC structures are similar, there are several major differences with important consequences that we emphasize in this revision with new figures and functional analysis of twenty-one additional mutants (Figures 3A-F, 4C, 5F-H, Figure 2—figure supplement 8, Figure 3—figure supplement 1). First, NKCC1 and KCC1 are dimers (albeit through different interfaces), while KCC4 is a monomer (Figure 3). We show that monomeric KCC4 is active and that its dimerization (through an interface analogous to NKCC1, but not KCC1) increases activity (Figure 3F, more below). Second, substantial rearrangement in the TM region between KCC4 and NKCC1 results in loss of a Na^+^ binding site in KCC4, providing a structural explanation for the difference in substrate specificity and stoichiometry between these transporters (Figure 5C). Third, two Cl^-^ ions are observed in the NKCC1 and KCC1 structures, while one Cl^-^ ion is observed bound to KCC4. Mutations around both sites impair transport activity, suggesting they are both involved in the KCC4 transport cycle (Figure 5F,G). Fourth, the extracellular domains in NKCC1 and KCC1/KCC4 are completely distinct: they are divergent in position within the transporter fold, in sequence, and in structure. Interactions unique to KCCs between the extracellular domain and TM region around the extracellular gate are functionally important (Figure 4C, more below).

The major difference is that KCC4 is a monomer and the others are dimers. This would be an opportunity to understand dimerization as a new aspect of these transporters. Unfortunately this difference is only addressed superficially in the discussion but without data or analysis. If you can provide evidence that the monomeric form of KCC4 is active, for example by ITC, it would be a valuable addition. If you could show that the ion transport or ion binding kinetics for the monomer and dimer are different, this would add considerable weight to your study.

We performed mutational analysis to assess whether dimerization through either of the distinct interfaces observed in KCC1 and NKCC1 is required for activity in cells (Figure 3). We found that monomeric KCC4 is active and that dimerization through an interface analogous to that observed in NKCC1 increases KCC4 activity (Figure 3F).

KCC1 dimerization predominantly involves protein-detergent interactions between TM regions and protein-protein interactions in an extracellular loop (Figure 3A,B). The extracellular loop is poorly conserved among KCCs (Figure 3E) and in KCC4 its structure is likely incompatible with dimerization due to steric clashes with a second protomer (Figure 3B). Consistently, introduction of three negative charges in the loop in KCC4 has no effect on transport activity. We conclude dimerization as observed in KCC1 is not functionally relevant for KCC4. In contrast, NKCC1 dimerization predominantly involves well conserved protein-protein interactions between CTDs (Figure 3C-E). Three separate pairs of mutations designed to disrupt an analogous interface in KCC4 result in a partial (~40%) reduction in activity. KCC4 retained activity even when the entire C-terminus was deleted (Figure 3F). Expression and folding were assessed for each mutant in the paper to ensure differences in these factors did not account for activity differences (Figure 3—figure supplement 1). We conclude that monomeric KCC4 is active and that dimerization increases KCC4 activity. This suggests the possibility of regulated dimerization in cells as a means for modulating transport activity in cells. Further exploring this possibility will be a focus of further study.

2) A related concern was that your study provides little in the way of insight into the regulation of the protein. While you propose that monomer-dimer transitions may occur during regulation of the protein in cells, no evidence is provided that this is actually the case.

Based on the structures and mutational studies, we can conclude that monomeric KCC4 is active and that dimerization (in a manner analogous to NKCC1) increases activity (Figure 3F). We believe a study of when this is utilized in cells or in vivo, while certainly very interesting, is outside the realm of this study. We reference a prior report suggesting regulated dimerization of KCC2 occurs in vivo and is associated with an increase in transport activity in conjunction with the neurodevelopmental excitatory-to-inhibitory GABA switch (Blaesse et al., 2006).

3) The fact that KCC4 is a monomer is surprising, given that the KCC1 structure, which is closely related, was determined as a dimer using a similar technique. Could the use of 1% DDM be responsible for disrupting the dimer during isolation? Native mass spectrometry might be a good way of analyzing this important aspect of KCC structure and regulation.

Differences in expression host cells or detergents used in the structures of CCCs do not appear to account for differences in oligomeric state between KCC4 and KCC1 (Figure 3—figure supplement 1). KCC4 and KCC1 displayed similar apparent size (judged by retention time during gel filtration) under all conditions tested (expression in HEK293T or SF9 cells and use of the three detergent combinations used in each CCC structure study (extraction: gel filtration detergents are DDM/CHS:DDM/CHS from KCC4, DDM/CHS:GDN from KCC1, and LMNG/CHS:digitonin from NKCC1).

Since KCC1 and KCC4 behave similarly in these experiments, why is there a difference in oligomeric state in the two structures? One possibility is that KCC1 and KCC4 dimerization interactions are weak and dimeric species are not significantly populated at the low concentrations used for FSEC or, in the case of KCC4, the μM concentrations used for cryo-EM. Regardless, our data suggests an analogous interface is not functionally important for KCC4 as mutation of the extracellular loop that forms the KCC1 dimer interface in a manner predicted to prevent dimerization has no effect on KCC4 function (Figure 3B,F).

In contrast, our data suggests dimerization of KCC4 through the CTDs (as observed in NKCC1) is functionally important and increases KCC4 activity (Figure 3D,F). Consistent with this mode of dimerization in KCCs, structures of KCC2 and KCC3 posted on bioRxiv show an apparently similar CTD-CTD interaction to that observed in NKCC1 (bioRxiv 2020.02.22.960815; doi: https://doi.org/10.1101/2020.02.22.960815). This study also used DDM/CHS and GDN detergents, so detergent differences cannot account for the oligomeric differences between different KCCs observed to date. We speculate differences in KCC oligomeric state and organization could have functional consequences and this will be a focus of future study.

4) A further point that would benefit from additional analysis is the role of the extracellular domain. You describe this domain as novel, but without investigating its function, either in cells or in vitro. You describe glycosylation, including what appears to be an interesting link between glycosylation at N312 and TM6A, which forms part of the transport machinery in APC proteins. Yet, the significance of this interaction is not clear. Is it an artifact or an important part of the mechanism?

We have provided additional mutational analysis of this domain as suggested. Glycosylation at N312 has been observed (at analogous sites) in KCCs in prior reports referenced. We show that glycosylation of N312 is functionally important, consistent with a link between the structure of this region and the extracellular gate around TM6A. KCC4 mutant N312Q results in a severe reduction in transport activity (Figure 4C) without obvious defects in folding or expression.

5) We also wondered about the extracellular gate, where you describe a salt bridge network that may function in a similar way to LeuT (and other APC transporters), but no analysis is provided. It would be good to characterize these interactions, for example through transport assays with variants at these sites and charge-swapped mutants. Such experiments would illuminate the mechanism of KCC4 and help to put the structure into a functional context.

We have added mutational analysis of the proposed gate between the TM region and extracellular domain (Figure 4C). Consistent with the importance of the hydrophobic and electrostatic interaction network we describe, mutation of key residues involved (K485D, K485E, F486A, and R140E) significantly reduce transport activity.

Reviewer #1:This paper describes the cryo-EM structure of KCC4, a K^+^/Cl^-^ cotransporter from *Mus musculus*. The structural work is well done, two residues in the ion-binding sites were mutated to demonstrate their importance to transport activity.My main concern is this:This is the third structure of this family of transporters. The other two, which were published earlier this year, are human KCC1 and DrNKCC1. The new structure in detail is nearly identical to the previous ones, including the ion binding sites. Some of the figures here are also similar.

Please see main concern #1 above. Some figures were purposefully made similar to facilitate simple comparisons with the NKCC1 paper.

The major difference is that KCC4 is a monomer and the others are dimers. This is an opportunity to understand an aspect of these transporters that until now was unknown. Unfortunately this difference is only addressed superficially in discussion but not with data and analysis. If the authors can provide evidence that the monomeric form of KCC4 is functional, then this work will make a good contribution towards our understanding of the architecture and function of KCCs.

Please see main concern #1-3. We provide evidence of functional monomeric KCC4 and that dimerization as observed in NKCC1 appears to increase activity, consistent with a proposed role for regulated oligomerization modulating transport activity.

Reviewer #2:[…]My main concern with the current manuscript is that apart from reporting the structure the study provides little else in the way of insight into the mechanism or regulation of the protein. I felt the manuscript was very descriptive in format but without any significant interpretation. The cross linking analysis appears to have provided little in the way of insight in to the oligomeric state for example. While the authors propose that monomer dimer transitions may occur during regulation of the protein in cells, no evidence is provided that this protein is capable of dimerization. This is rather odd, especially given that the KCC1 structure, which is closely related, was determined as a dimer using a similar technique. I wondered whether the use of 1% DDM might be responsible for disrupting the dimer interaction? I also wondered whether native mass spectrometry might be a better way of analyzing this important aspect of KCC regulation.

Please see Main concern #1-3 above. We provide evidence that dimerization as observed in KCC1 is irrelevant for KCC4 function, for functional monomeric KCC4, and for dimerization as observed in NKCC1 increasing transport activity. We agree understanding when in cells regulated oligomerization could occur is an interesting and important next step, but would prefer to explore this broad question in future work.

Another area of the current study that I found frustrating was the lack of analysis on the role of the extracellular domain. For example, the authors describe the novel extracellular domain, but do not investigate its function, either in cells or in vitro. They describe the glycosylation observed, including what appears to be an interesting link between glycosylation at N312 and TM6A, which forms part of the transport machinery in APC proteins. Yet, the study leaves the reader wondering what the significance of this interaction is. Is it an artifact or an important part of the mechanism? I wondered the same when reading the section that describes the extracellular gate. Here the authors describe a set of salt bridge networks that appear to function similar to LeuT (and other APC transporters), but no analysis is provided. I would have liked to see an attempt made to functionally characterize these interactions, either through transport assays with variants at these sites and charge swapped mutants. Such experiments would greatly illuminate the mechanism of KCC4 and help to put the structure in the context of the function. The same could be said of the bifurcated pathway – artifact or interesting mechanistic observation. Here mutations and molecular dynamics could shed more insight.

Please see main concern #4 and #5. We provide evidence that both glycosylation and the electrostatic and hydrophobic interaction network that structurally couples the extracellular domain to the TM region and the outer gate are functionally important.

In relation to the comparison between KCC4 and NKCC4, the latter protein binds to Na^+^ in addition to K^+^ and Cl^-^, I was interested to know what compensatory interactions are made that enables KCC4 to dispense with sodium binding. In the APC transporters that move protons for example, the second Na^+^ site is substituted with a lysine. I felt this part of the analysis was also less in depth that warranted.

The differences are shown in Figure 5C and detailed in the text. Rather than introduction of a compensatory charged residue like lysine, loss of the Na^+^ site involves large scale helical shifts and mutation of coordinating residues to more broadly change the chemical nature of the Na^+^ site.

Reviewer #3:[…]1) KCC4 induced Tl^+^ uptake activity has been demonstrated in intact cells. However, dependence on the symported ion-Cl has only been illustrated. Please, clarify why?

One reason is that only a narrow range of [Cl^-^]_ext_ is accessible using the Tl^+^ assay due to the very low solubility of TlCl salt that interferes with the assay. We agree this will be an interesting avenue to explore in future work with a different assay.

2) Although, the Tl^+^ uptake due to KCC4 activity was nicely determine in intact cells the structure was determined in nanodiscs. Therefore, there is no indication that the revealed structure is the native functional protein structure. Activity of the protein in nanodiscs is missing and this can be accomplished by testing binding of K^+^/Cl^-^. Using ITC is a simple option.

We believe a thermodynamic characterization of ion binding, while certainly interesting, is best suited for future work. We think the structure observed is functionally relevant for the following reasons. First, we observe ions bound in the structure and mutations at these sites disrupt transport. Second, residues predicted from the structure to form part of a previously unseen extracellular gate interaction and a unique connection between a glycosylation site and the gate were shown to be functionally important. Third, our mutational data at putative dimerization sites is consistent with functional monomeric KCC4. Fourth, five distinct CCCs have now been structurally captured in an overall similar conformation.

3) Structure based mutants support the proposed binding sites and have raised interesting discussion regarding ion/substrate selectivity in this group of transporters. The authors also suggest that amino acid residues residing in the protein's extracellular domain form an outward gate of the transporter. This interesting suggestion has to be explored by analyzing mutations at least of the residues that are localized outside the membrane part of the transporter.

Please see main concern #4 and 5.